# Efficacy of SARS-CoV-2 Vaccination in Dialysis Patients: Epidemiological Analysis and Evaluation of the Clinical Progress

**DOI:** 10.3390/jcm11164723

**Published:** 2022-08-12

**Authors:** Giovanni Mosconi, Michela Fantini, Matteo Righini, Marta Flachi, Simona Semprini, Lilio Hu, Francesca Chiappo, Barbara Veterani, Katia Ambri, Franca Ferrini, Catia Milanesi, Antonio Giudicissi, Gaetano La Manna, Angelo Rigotti, Andrea Buscaroli, Vittorio Sambri, Maria Cappuccilli

**Affiliations:** 1Nephrology and Dialysis Unit, AUSL Romagna Morgagni-Pierantoni Hospital, 47121 Forli, FC, Italy; 2Local Healthcare Authority of Romagna (AUSL Romagna), 48121 Ravenna, RA, Italy; 3Nephrology and Dialysis Unit, AUSL Romagna S. Maria delle Croci Hospital, 48121 Ravenna, RA, Italy; 4Nephrology and Dialysis Unit, AUSL Romagna Infermi Hospital, 47923 Rimini, RN, Italy; 5Unit of Microbiology, AUSL Romagna Laboratory, 47023 Pievesestina, FC, Italy; 6Nephrology Dialysis and Renal Transplant Unit, IRCCS-Azienda Ospedaliero-Universitaria di Bologna, Alma Mater Studiorum University of Bologna, 40138 Bologna, BO, Italy

**Keywords:** COVID-19, COVID-19 vaccination, hemodialysis, immunodepressed patients, mRNA vaccines, SARS-CoV-2 infection

## Abstract

This study investigated the impact of the fourth COVID-19 pandemic wave on dialysis patients of Romagna territory, assessing the associations of vaccination status with infection risk, clinical severity and mortality. From November 2021 to February 2022, an epidemiological search was conducted on 829 patients under dialysis treatment for at least one month. The data were then analyzed with reference to the general population of the same area. A temporal comparison was also carried out with the previous pandemic waves (from March 2020 to October 2021). The epidemiological evolution over time in the dialysis population and in Romagna citizens replicated the global trend, as the peak of the fourth wave corresponded to the time of maximum diffusion of omicron variant (B.1.1.529). Of 771 prevalent dialysis patients at the beginning of the study, 109 (14.1%) contracted SARS-CoV-2 infection during the 4-month observation period. Vaccine adherence in the dialysis population of the reference area was above 95%. Compared to fully or partially vaccinated subjects, the unvaccinated ones showed a significantly higher proportion of infections (12.5% vs. 27.0% *p* = 0.0341), a more frequent need for hospitalization (22.2% vs. 50.0%) and a 3.3-fold increased mortality risk. These findings confirm the effectiveness of COVID-19 vaccines in keeping infectious risk under control and ameliorating clinical outcomes in immunocompromised patients.

## 1. Introduction

The coronavirus disease 2019 (COVID-19) pandemic still has a dramatic impact worldwide, even in high-income countries that are experiencing heavy rebounds in health care systems, lifestyles, social habits and interpersonal relationships. As of 25 May 2022, the WHO reported over 524 million confirmed cases of SARS-CoV-2 infection and above 6 million deaths worldwide (https://covid19.who.int/, accessed on 25 May 2022). In Italy, the National Institute of Health (Istituto Superiore di Sanità) reported a cumulative incidence since the onset of the COVID-19 pandemic of 28,538 laboratory-confirmed cases per 100,000 inhabitants (https://www.epicentro.iss.it/coronavirus/aggiornamenti, lastly released national update of 18 May 2022).

Some fragile categories have been burdened by significantly increased morbidity and mortality from COVID-19 compared to the general population. Among patients with chronic kidney disease (CKD) requiring dialysis treatment, an overwhelming proportion of active cases and deaths has been described since the early phases of the COVID-19 outbreak in 2020, as well as during the successive pandemic waves [1,2,3]. Several studies have proven that patients under chronic kidney replacement therapy show COVID-19 mortality rates ranging between 20% and 30% [4]. Moreover, the reported SARS-CoV-2 seroprevalence is up to 40% in patients on dialysis, even in the presence of the asymptomatic disease [5]. The elevated infectious risk in these patients can be partly explained by the need for repeated access to hospital/health care facilities, transport from home to the dialysis center and prolonged treatment in contact with other patients and staff [6,7]. The common presence of cardiovascular comorbidity, overweight, diabetes, associated with inflammation, accelerated immunosenescence and ultimately age-related immune dysfunction is another underlying cause for the unfavorable outcomes [8].

In such weak patients, the link between immune responsiveness and clinical presentation of COVID-19 is unpredictable and might be largely influenced by genetic factors, such as polymorphisms in ACE1/ACE2 or other genes of the renin-angiotensin system (RAS) [9,10]. Since the onset of the pandemic, dialysis centers have implemented preventive actions in an effort to reduce transmission: triage before entering the dialysis room, separation of access routes between COVID and non-COVID patients, isolation of positive patients or those with previous close contact, careful clinical monitoring, screening tests, use of masks, hand washing, and personalized transportation [11,12,13].

In addition to the important evolution of technical and scientific knowledge, including the progression of pharmacological research on novel antiviral agents [14], at present, mass vaccination represents the most powerful pandemic strategy to control COVID-19 [15,16]. Chronic dialysis was identified among the medically frail conditions to prioritize access to COVID-19 vaccination [17], but current data on the coverage and durability of vaccine-induced immune protection in uremic patients are inconsistent. Immunodepressed categories indeed commonly show heterogeneous behaviors towards SARS-CoV-2 infection and vaccine responsiveness [18,19,20]. In our experience, we observed a prolonged viral clearance in dialysis patients who recovered from COVID-19 in front of a sufficient antibody response, which, however, tends to wane rapidly over time [21,22]. Compared to immunocompetent subjects with normal renal function, data on the responsiveness of dialysis patients toward COVID-19 vaccination are elusive [23].

This study was aimed at investigating the impact of the fourth COVID-19 pandemic wave in a population of dialysis patients, with special attention to the efficacy of vaccination on the viral transmission, clinical severity, need for hospitalization and mortality due to SARS-CoV-2 infection.

## 2. Materials and Methods

### 2.1. Study Design and Population

This is a multicenter observational study to assess the possible relationship of COVID-19 vaccination status with viral transmission, disease virulence and mortality during the period of the fourth pandemic wave. From November 2021 to February 2022, an epidemiological analysis on COVID-19 was conducted on the citizens of the northern Italian territory of Romagna (1,114,447 inhabitants as of 1 January 2022) and on patients under dialysis treatment for at least one month at the Nephrology and Dialysis Units of the local health authorities in the same area (Forlì-Cesena, Rimini and Ravenna).

The study period was chosen according to the detection in autumn 2021 of a new rise of COVID-19 infections until the pandemic peak prior to the progressive decline at the end of February 2022. The COVID-19 situation for the general population of Romagna was derived from the weekly epidemiological regional update.

Since the beginning of the pandemic outbreak in Italy in March 2020, general measures to keep virus transmission under control in dialysis facilities have been applied. Then, when COVID-19 vaccines were introduced, the dialysis patients were recommended to start vaccination with priority from March 2021. The dialysis units took charge of also organizing the administration of booster doses in line with the national directives. All the dialysis patients received mRNA vaccines (BNT162b2 vaccine, Comirnaty, Pfizer-BioNTech or mRNA-1273 vaccine, Spikevax, Moderna). The second dose was scheduled 21–28 days after the first dose, depending on guidelines for spacing vaccine shots. The rollout of the booster dose began in September 2021, at least 4 months after the second vaccine injection or an eventual SARS-CoV-2 infection. All the patients starting dialysis treatment were recommended to receive a full vaccination series.

The diagnosis of SARS-CoV-2 infection was based on nasopharyngeal swab positivity regardless of clinical symptoms. Disease severity, need for hospitalization, outcomes and time between diagnosis and recovery were evaluated throughout the observation period. The extent of clinical involvement was assessed by subdividing the infected patients into: (a) asymptomatic/paucisymptomatic (presence of one or more mild symptoms not requiring hospitalization); (b) symptomatic, requiring hospitalization.

The mortality rate was evaluated as COVID 19-related death in the course of infection. A comparative analysis was also made between different dialysis modalities, specifically hemodialysis at the hospital or in outpatient dialysis facilities versus home peritoneal dialysis.

At the time of enrolment, the status of each participant with respect to vaccine-induced immune coverage was registered as follows: (i) fully vaccinated in case of a recent complete three-dose vaccine or less than three doses and a recent infection; (ii) partially vaccinated in case of non-recent last vaccine dose administration; (iii) no vaccination. The term “recent” means a time interval below 120 days from that last dose vaccine administration to the end of the study for the non-infected patients or to the time of positivity detection for those infected during the observation period. The vaccination program continued during the observation period. Any changes from the vaccination schedule or adherence were recorded and analyzed for the possible rebounds in viral transmission, disease virulence and death risk. In the patients who experienced the less severe form of the infection, home therapy was based on monoclonal antibodies modulated in relation to the changing pattern of circulating SARS-CoV-2 strains (Casirivimab or Imdevimab in the presence of higher delta prevalence, Sotrovimal during omicron prevalence), supported by oral steroid therapy in 6 cases and antibiotic treatment in 3 cases. The symptomatic patients requiring hospitalization received medical interventions and supportive oxygen therapy with non-invasive or invasive ventilation according to the degree of respiratory distress and to current guidelines.

The overall analysis was also focused on the time evolution of the COVID-19 pandemic, evaluating the temporal variations between the first three waves (20-month period from March 2020 to October 2021) and the fourth wave (4-month period from November 2021 to February 2022).

This study was conducted according to the guidelines of the Declaration of Helsinki and approved by the Institutional Ethics Committee “Comitato Etico della Romagna, CEROM” (code INCoV19ID, date of approval 11 December 2020). Informed consent was obtained from all subjects involved in the study, and patients’ data were fully anonymized.

### 2.2. Statistical Analysis

Continuous variables are presented as means ± standard deviation (SD) if normally distributed and as median with interquartile range (IQR) if non-normally distributed. Categorical variables are presented as absolute numbers and percentages. Continuous variables were analyzed through the Student’s *t*-test or the non-parametric Mann–Whitney U or two-way analysis of variance (ANOVA) followed by Tukey’s *t*-test, as appropriate. A Chi-square test with a contingency table was used to compare proportions between categorical variables. The possible associations of vaccination status with mortality risk were evaluated through odds ratio (OR) and 95% confidence intervals (95% CI). A *p*-value below 0.05 was considered significant, and all the statistical analyses were performed using the statistical package for the social sciences (SPSS™ for Windows Software Package, version 9.0.1; Chicago, IL, USA).

## 3. Results

### 3.1. Characteristics of the Population of Dialysis Patients

A total of 829 patients treated with dialysis therapy for at least one month during the period from 1 November 2021 to 28 February 2022 were investigated. At the beginning of the study, the prevalent dialysis population consisted of 771 patients, 721 (93.5%) under hemodialysis and 50 (6.5%) under peritoneal dialysis. Over the 4-month observation period, 58 patients started hemodialysis (*n* = 50) or peritoneal dialysis (*n* = 8), and 52 left the program due to death, kidney transplantation or moving to another dialysis center. At the end of the study, the prevalent dialysis population consisted of a total of 777 patients, 722 (92.9%) of them under hemodialysis and 55 (7.1%) under peritoneal dialysis.

The main features of this cohort divided according to dialysis modality are reported in Table 1. The two groups were comparable for gender distributions, weight and body mass index (BMI), the occurrence of hypertension, glomerulonephritis, polycystic kidney disease, IgA nephropathy, hereditary renal diseases, vascular nephropathy and undiagnosed renal disease as primary causes of kidney function failure. On the other hand, patients in home peritoneal dialysis were younger, had a shorter dialysis vintage and showed an inferior frequency of cardiovascular disease (CVD), diabetes, diabetic nephropathy and interstitial nephritis as primary renal diseases.

### 3.2. Epidemiology of COVID-19 in the General Population and in Dialysis Patients

#### 3.2.1. General Population

Retrospective data of Romagna inhabitants (*n* = 1,122,114, as of 1 January 2021) from the beginning of the COVID-19 pandemic outbreak in March 2020 to October 2021 revealed a 10.5% proportion of infections, with a mortality rate of 2.6% (calculated as the number of COVID-19-related deaths per cumulative cases of the same period).

The epidemiological analysis on the same territory (1,114,447 inhabitants, as of 1 January 2022) for the fourth pandemic wave (the 4-month period between November 2021 and February 2022) found a remarkable increase in the proportion of infections to 20.8% and a drop in mortality rate to 0.3%. The region tracker showed a peak in positivity detection between the 2nd and the 3rd week of January 2022, largely comparable in the different territorial districts (Figure 1).

In our reference area, the epidemiological course over time was consistent with the global trend, as the zenith of the fourth wave corresponded to the time of maximum diffusion of the omicron variant (B.1.1.529). Sequencing analysis of viral genome performed on a weekly basis on random infected patients confirmed that in the second half of January, the omicron variant almost completely replaced the delta (B.1.617.2) (Table 2).

#### 3.2.2. Dialysis Patients

In the dialysis population of the reference area, retrospective data analysis of the 20-month period prior to the fourth wave (March 2020–October 2021) revealed that 136 out of 1108 dialysis patients (12.3%) treated in our Romagna dialysis units were infected. When considering the dialysis modality, the first three pandemic waves resulted in a nearly significant (*p* = 0.0567) higher susceptibility associated with hemodialysis treatment (132 cases out of 1027, 12.8%) compared to peritoneal dialysis (4 cases out of 81, 4.9%). With respect to the general population, dialysis patients showed a tendentially increased percentage of infections (10.5% vs. 12.3%, *p* = 0.054).

Among the patients under dialysis treatment for more than one month in the period November 2021−February 2022, 109 out of 771 patients contracted SARS-CoV-2, showing a significantly lower proportion of infections compared to the Romagna population (14.1% vs. 20.8%, *p* < 0.001). Concerning the epidemiological situation over time, our dialysis patients mirrored the trend of the general population, as 94 out of 109 (86.2%) positive detections of SARS-CoV-2 were found between 4 January 2022 and 28 February 2022, with similar patterns in the various participating dialysis units (single-center data not shown). The fourth wave accounted for 44.5% of the cases in the dialysis population of the Romagna area (109 out of 245 total SARS-CoV-2 infections since the beginning of the pandemic). Although the viral infection was again more common in hemodialysis patients than in those under peritoneal dialysis, the limited sample size prevented differences between groups from reaching statistical significance (13.6% vs. 7.0%, *p* = 0.2019).

The weekly new cases of COVID-19 in the general population and in dialysis patients are detailed in Figure 2.

### 3.3. Vaccination Status and Infection of COVID-19 in Dialysis Patients

In our dialysis units, vaccine adherence was higher than that of the general population. In the overall population of 829 patients followed up between November 2021 and February 2022, 790 (95.3%) received SARS-CoV-2 mRNA-based vaccination. In line with the national and regional indications, we evaluated the immune coverage based on the number of administered doses and the time interval since the last dose. Accordingly, among the 790 vaccinated participants, 337 were fully covered (complete three-dose vaccine or less than three doses plus infection in the 120 days prior to the study), and 453 were partially covered (last vaccine dose administration less recent than 120 days). Eighteen dialysis patients who were unvaccinated at the beginning of the study started the vaccination cycle during the 4-month observation period. The indications for spacing vaccine shots were mostly followed, although in a few cases, there were some time shifts due to acute clinical problems or other intercurrent necessities related to the dialysis procedure itself. A total of 39 (4.7%) patients remained unvaccinated.

A significantly higher frequency of SARS-CoV-2 infections was noticed in non-vaccinated subjects (10/37, 27.0%) compared to those who had a full or partial immune coverage (99/792; chi-square: 27.0% vs. 12.5%, *p* = 0.0341).

Table 3 compares the main features of the 109 dialysis patients who contracted SARS-CoV-2 infection during the study period, divided according to the vaccination status. The 10 unvaccinated subjects were significantly younger compared to the other groups, regardless of the number and time of dose administration. The need for hospitalization was more frequent in unvaccinated patients (50%) compared to those with a full or partial vaccine-induced immune coverage (21.3% and 25.0%, respectively; nearly significant *p* = 0.071).

### 3.4. Mortality

Between November 2021 and February 2022, the mortality rate was 5.3% (4/75) in fully vaccinated dialysis patients, 8.3% (2/24) in partially vaccinated and 20% (2/10) in non-vaccinated. The patients without any vaccine dose had a 3.3-fold increased mortality risk compared to those who were vaccinated, regardless of time from the last dose administration [OR: 3.30; 95% CI: 0.586 to 18.566; *p* = 0.175].

Among the patients who were fully or partially vaccinated, three died in the course of COVID-19 positivity from causes unrelated to respiratory failure due to exacerbation of the pre-existing severe cardiovascular disease.

When considering the mortality rate in the same population censored for these three patients, the mortality risk rose to a 6.6-fold higher in the unvaccinated dialysis patients compared to the fully or partly vaccinated [OR: 6.60; 95% CI: 0.983 to 44.29; *p* = 0.0520].

## 4. Discussion

The COVID pandemic resulted in significant changes in the care management of patients requiring dialysis treatment. At the beginning of the pandemic outbreak in 2020, the clinical repercussions were dramatic, especially in older and comorbid patients. Mortalities from 30% up to 40% were reported in dialysis-dependent patients [3,24,25]. Different logistical and organizational interventions have been useful in preventing the spread of the infection, thanks to the extensive employment of financial and professional resources. Our work confirms that the dialysis population has shown an elevated susceptibility to COVID-19 since the beginning of the pandemic. Between March 2020 and November 2021, the overall incidence of SARS-CoV-2 infection in patients followed in our dialysis units was 12.3%, higher with hemodialysis treatment (12.8%) compared to peritoneal dialysis (4.9%).

Data analysis was complicated in a population with a fast turnover: at the end of the observation period, 12.6% of our patients still under dialysis had a recent infection contracted during the 4-month observations period.

The onset of new highly diffusive variants of concern resulted in an unprecedented rapid growth in SARS-CoV-2 cases, especially after December 2021 [26,27]. In the report of 14 December 2021, the World Health Organization described the omicron variant as the faster transmissible viral strain than any previously detected one. Up to now, vaccination has indeed demonstrated highly significant efficacy in lowering the rate of COVID-19 severe illnesses and deaths in the general population [28,29,30,31]. Our findings confirm that the institution of population-wide vaccination programs started in Europe at the end of the year 2020 also provided major benefits in terms of COVID-19 mortality and hospitalizations in weak patients, including those on dialysis [32,33]. Over the different national health systems, vaccination strategies, in terms of the number of doses and administration time schedules, were progressively adapted, in the general population as well as in vulnerable patients, to the emerging scientific evidence and pandemic evolution along with the appearance of new viral strains [34,35].

During the first three pandemic waves in our area, SARS-CoV-2 infection was found with higher frequency in dialysis patients compared to the general population, in line with national [24,25] and international data (https://ourworldindata.org/covid-cases, accessed on 25 May 2022).

A reversed trend was found for the fourth wave, as the percentage of the period November 2021–February 2022 was 14.1% in dialysis patients and 20.8% in the general population. This might be explained by two main factors: on the one hand, the priority and higher adherence to vaccination (above 95%) of dialysis patients as a weak category; on the other, the different diffusion of omicron variants by age groups. Epidemiological data showed that this variant mostly affected younger subjects (https://www.epicentro.iss.it/coronavirus/aggiornamenti, report of 18 May 2022), that are underrepresented in the dialysis population. In our reference area, from November 2021 to February 2022, the mean age of infected subjects was 37.4 years among the citizens of Romagna and 69 years for dialysis patients. Moreover, the overall striking decrease in mortality rate seen in the fourth wave with respect to the previous period (from 2.6% to 0.3%) might reflect the synergistic effects of vaccine-induced protection and the spread of the omicron variant that seems to cause a milder disease due to its reduced ability to infect lung cells [36].

COVID-19 vaccination is strongly recommended and prioritized for frail subjects with chronic conditions. The clinical characteristics of patients undergoing renal replacement therapy and the particularly elevated mortality rate found in the first year of the pandemic led the health authorities to include this population in a priority vaccination program, including both primary series and booster doses [37]. However, given the uremia-related immune dysfunction, data on responsiveness to COVID-19 vaccines in dialysis patients are inconclusive, similarly to other vaccinations (i.e., inactivated influenza, pneumococcal and hepatitis B vaccines) [23,37,38].

Previous data from our group on patients with impaired renal function described a delayed viral clearance and an antibody response that tends to decline more rapidly than in the general population [21,22]. In dialysis patients, there is limited awareness of the cellular immunity induced by vaccination [39,40], and comparative studies on incidence, clinical course and outcomes between vaccinated and unvaccinated patients are lacking [41]. In our dialysis facilities, we could observe a satisfying adherence to vaccination schedules, higher than the one recorded in the general Italian population. However, there was some variability in terms of distribution and timing of vaccine schedules, feasibly due to concurrent clinical problems delaying programmed shots, an elevated patient turnover and some initial hesitancy from the patients. Our experience further emphasizes the essential role of thorough counseling operated by health personnel, especially in a clinically and psychologically fragile population whose choices are often conditioned by families and caregivers [42]. The period analyzed in this study was that of the fourth pandemic wave in Italy, characterized by the appearance of the highly contagious omicron variant that quickly replaced the delta, as confirmed by sequencing analysis on infected inhabitants of the Romagna region, as well as by national and international data [43]. The most remarkable finding was the lower rate of COVID-19-related complications, severe symptoms, hospitalizations and deaths observed after vaccine introduction in comparison with the initial pandemic phases, even in vulnerable subjects with advanced age and pre-existing chronic conditions. In addition to the crucial role of vaccines, other factors have made an important contribution to the control of the initially distasteful effects of COVID-19 in dialysis patients. The experience acquired over time since the first pandemic phases paved the way for the collaboration between interdisciplinary teams [44], the adoption of targeted measures (dedicated rooms, dedicated pathways) [45] and the development of earlier and more sensitive diagnostic tests [46,47], together with the pharmacological research on novel therapeutic agents [48,49].

Although Italy has been heavily burdened by the pandemic, especially in the beginning period [50], the healthcare system was able to cope with the huge numbers of patients needing critical assistance, including those with significant comorbidities [51].

In our population, there was no evidence of clusters within any of the dialysis facilities, and in about 90% of the cases, the transmission feasibly occurred within the family, although tracking data should be interpreted with caution for the most dramatic periods of COVID-19 epidemic peak. However, in spite of all the precautions taken to contain the spread of viral infection [24,25], hemodialysis facilities remain an environment at high risk of contagion, and the different transmission rate found between patients on hemodialysis and those on home peritoneal dialysis confirms this view.

To our knowledge, this study is the first comparison within a fragile dialysis-dependent population on the impact of SARS-CoV-2 infection in vaccinated and unvaccinated subjects. Although the mostly described benefits of vaccines lie in their ability to reduce COVID-related hospitalizations and mortality [52,53], we could also observe the satisfying effects of contagion control in our dialysis patients. In particular, our data found lower rates of infections, hospitalizations and deaths in the fully or partially vaccinated subjects compared to the unvaccinated, a point that is still under investigation for the general population [54,55,56]. Nevertheless, consistent with data from non-renal immunocompetent individuals, the vaccination status of our dialysis patients has significant implications for mitigating the most detrimental effects of the virus in a frail population. It can be stated that the initiation of mass vaccination campaigns reduced clinical complications and mortality rates with respect to the previous pandemic waves [1,2,3]. Moreover, our data show that 50% of the unvaccinated patients contracted the severe form of the disease with the necessity of hospitalization and respiratory support, while the frequency of hospital admissions was around 22% in those who received a vaccination, regardless of the status of full or partial immune coverage. Likewise, the mortality rate was 20% in unvaccinated subjects and 6% in fully or partly vaccinated ones (5.3% and 8.3%, respectively). The status of vaccine-induced immune coverage was correlated with mortality rate, as we noticed a remarkably increased death risk in unvaccinated dialysis patients.

It is worth mentioning that while the two unvaccinated patients died of virus-triggered severe respiratory complications, this is not the case of three patients among vaccinated patients who died of a previous major cardiovascular event in the course of SARS-CoV-2 positivity. The challenging issue of how to differentiate and categorize deaths from or with coronavirus regarded half of our patients, but only in the vaccinated group. Considering the dates of deaths in relation to viral genome sequencing analysis over time, we can presume that one patient in the unvaccinated group (*n* = 10) and one in the fully vaccinated group (*n* = 75) died of severe pulmonary distress at the beginning of January 2022 feasibly contracted the delta variant. This strain is known to be less transmissible but more likely to escape vaccine protection and produce symptomatic and risky infections than omicron [57].

Unfortunately, we did not perform genetic sequencing in our dialysis cohort, and this represents one current limitation of this study, together with a lack of data on the measurement of antibody titers in response to vaccination, vaccine-induced immune responses prior to the infection (e.g., antibody titers, neutralizing antibodies, cellular immunity). Indeed, our analysis was conducted on a large population consisting of all the patients following accessing the dialysis facilities in Romagna territory, an area that covers a population of 1,114,447 (estimate as of 1 January 2022). Thus, the small sample size of the non-vaccinated subjects, which can be seen as a drawback in terms of statistical robustness, further confirms the satisfactory adherence to vaccination programs in our dialysis facilities. Nevertheless, a very recent study from our research group reported an overall satisfying level of protection after a two-dose vaccination cycle with mRNA vaccines in dialysis patients, and renal transplant recipients recovered from COVID-19, with a maximum peak at 3 months after the second dose. Although these data were described in a limited number of patients, we were able to observe good responsiveness to COVID-19 vaccination even in those patients with immune dysfunction, corroborating the notion the vaccination can provide stronger protection against re-infection and COVID severe illness with respect to the infection-induced immunity, but the combined effect of both can trigger the strongest response [58].

One of the strengths of our work was the availability of viral genome sequencing in the reference area on a weekly basis. It is highly probable that in our cohort of dialysis patients, the distribution of viral strains over time replicated the pattern found in infected subjects submitted to viral genome analysis every week in the Romagna territory.

The number of cases is another important factor that deserves some mention. In our patients who undergo a dialysis session on average three times a week, the tracking of the infection is carried out with particular attention, including pre-dialysis triage actions and frequent contact with the family environment. This has allowed good infection monitoring despite the difficulties present in an environment at high transmission risk. The general containment policies, management strategies and therapeutic interventions were homogeneous throughout the territory and in all the participating centers.

In spite of the relatively short observation period, different pharmacological strategies have been adapted to the evolution of the pandemic, using newly available monoclonal antibodies targeted to specific viral strains (casirivimab/imdevimab effective against delta, sotrovimab effective against omicron variant).

## 5. Conclusions

Patients undergoing dialysis treatment represent a high-risk category of SARS-CoV-2 transmission and are more susceptible to unfavorable clinical outcomes due to cardiovascular comorbidities and uremia-related immune dysfunction. Our findings suggest that even in immunocompromised patients in whom the ability to develop antibody protection is not well defined, vaccination has a positive effect on the clinical severity of the infection, the need for hospitalization and mortality. In the medium term, as in the case of the general population, it will be necessary to try to adapt resources to the ever-changing epidemiological trend in view of a hopeful transition from pandemic to endemic. However, the approaches to frail patients should, in any case, be prudent and gradual while maintaining a strict monitoring level. A primary task of the medical and nursing team in charge of chronically weak patients is to provide constant counseling and to encourage adherence to vaccination programs and to other preventive measures in relation to continually evolving scientific knowledge.

## Figures and Tables

**Figure 1 jcm-11-04723-f001:**
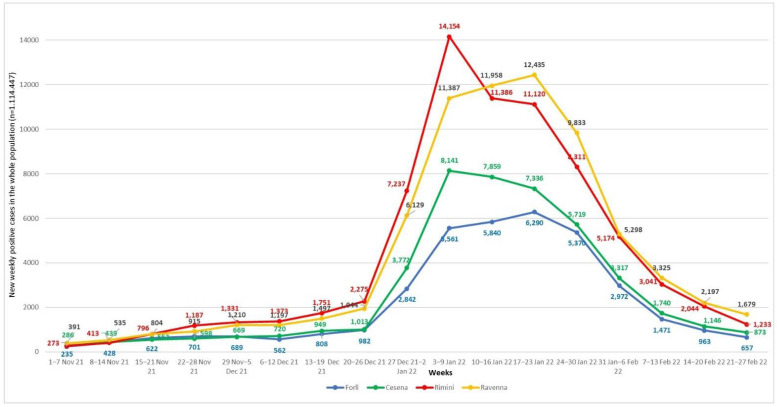
Weekly confirmed COVID-19 cases were divided according to Romagna territorial districts (Forlì, Cesena, Rimini and Ravenna) from 1 November 2021 to 28 February 2022.

**Figure 2 jcm-11-04723-f002:**
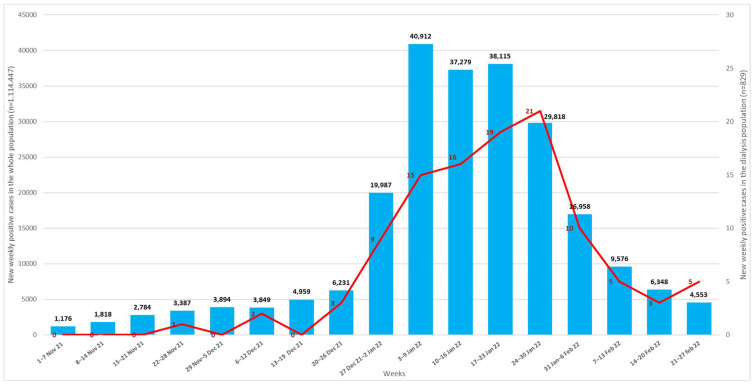
Weekly confirmed COVID-19 new cases from 1 November 2021 to 28 February 2022 in the general population (blue bars) and in dialysis patients (red line) of Romagna territory.

**Table 1 jcm-11-04723-t001:** Demographic and clinical variables in total hemodialysis and peritoneal dialysis population patients were followed during the 4-mont period (from November 2021 to February 2022) at the Nephrology and Dialysis Units of the local health authority of Romagna (Forlì-Cesena, Ravenna and Rimini). Continuous variables are presented as means ± standard deviation (SD) in normally distributed and as median with interquartile range (IQR) in square brackets if non-normally distributed, and nominal variables are presented as percentage and absolute numbers in brackets. Significant *p*-values are in bold.

	Total Dialysis Population (*n* = 829)	A. Hemodialysis (*n* = 772)	B. Peritoneal Dialysis (*n* = 57)	*p* Value (A vs. B)
Gender, M (*n*, %)	581 (70.1%)	539 (69.8%)	42 (73.7%)	0.798 ^a^
Age (years)	69.2 ± 13.5	69.6 ± 13.6	63.6 ± 14.9	**0.00108** ^b^
Dialysis vintage (months)	37.0 [15.4–75.9]	40.0 [15.7–81.2]	20.9 [9.9–32.1]	**<0.0001** ^c^
Body weight (kg)	70.2 ± 16.6	69.9 ± 24.9	25.0 ± 4.1	0.158 ^b^
BMI (kg/m^2^)	24.9 ± 3.4	24.9 ± 3.3	63.6 ± 14.9	0.862 ^b^
CVD (*n*, %)	428 (51.6%)	419 (54.2%)	9 (15.8%)	**<0.0001** ^c^
Diabetes (*n*, %)	206 (24.8%)	202 (26.2%)	4 (7.0%)	**0.0071** ^c^
Hypertension (*n*, %)	356 (42.9%)	322 (41.7%)	34 (59.6%)	0.113 ^c^
Previous renal transplant (*n*, %)	80 (9.6%)	80 (10.4%)	0 (0%)	/
Primary disease (*n*, %)				
Glomerulonephritis	44 (5.3%)	42 (5.4%)	2 (3.2%)	0.548 ^c^
PKD	62 (7.5%)	61 (7.9%)	1 (1.8%)	0.105 ^c^
Hypertension	180 (21.7%)	160 (20.7%)	20 (35.1%)	0.052 ^c^
Diabetic nephropathy	141 (17.0%)	139 (18.0%)	2 (3.5%)	**0.0121** ^c^
IgA nephropathy	13 (1.6%)	11 (1.4%)	2 (3.5%)	0.233 ^c^
Hereditary nephropathy	52 (6.3%)	50 (6.5%)	2 (3.5%)	0.397 ^c^
Vascular nephropathy	48 (5.8%)	45 (5.8%)	3 (5.3%)	0.867 ^c^
Interstitial nephritis	18 (2.2%)	13 (1.7%)	5 (8.8%)	**<0.0001** ^c^
Undiagnosed	271 (32.7%)	251 (32.5%)	20 (35.1%)	0.778 ^c^

BMI, body mass index; CVD, cardiovascular disease; PKD, polycystic kidney disease. ^a^ chi-square; ^b^ Student’s *t*-test; ^c^ Mann–Whitney U test.

**Table 2 jcm-11-04723-t002:** Weekly percentage distributions of SARS-CoV-2 variants identified by viral genome sequencing in new positive samples of the Romagna territory.

Week	Delta	Omicron
1−7 November 2021	100%	0%
8−14 November 2021	100%	0%
15−21 November 2021	100%	0%
22−28 November 2021	100%	0%
29 November−5 December 2021	100%	0%
6−12 December 2021	100%	0%
13−19 December 2021	100%	0%
20−26 December 2021	93%	7%
27 December 2021−2 January 2022	64%	36%
3−9 January 2022	31%	69%
10−16 January 2022	20%	80%
17−23 January 2022	0%	100%
24−30 January 2022	5%	95%
31 January−6 February 2022	6%	94%
7−13 February 2022	3%	97%
14−20 February 2022	0%	100%
21−27 February 2022	0%	100%

**Table 3 jcm-11-04723-t003:** Comparison of demographic, clinical and COVID-related variables in dialysis patients infected during the 4-mont period (from November 2021 to February 2022), grouped according to the vaccination status. Continuous variables are presented as means ± standard deviation (SD) in normally distributed and as median with interquartile range (IQR) in square brackets if non-normally distributed, and nominal variables are presented as percentage and absolute numbers in brackets. Significant *p*-values are in bold.

Total Dialysis Population Infected with SARS-CoV-2 in the Observation Period (*n* = 109)	A. Fully Vaccinated (*n* = 75)	B. Partially Vaccinated (*n* = 24)	C. Non Vaccinated (*n* = 10)	*p*-Value
Gender (M)	55 (73.3%)	21 (87.5%)	6 (60.0%)	0.791 ^a^
Age (years)	69.4 ± 13.4	66.2 ± 14.4	49.9 ± 17.0	**0.0003** ^b,§^
Type of dialysis				
Hemodialysis Peritoneal dialysis	72 (96.0%) 3 (4.0%)	23 (95.8%) 1 (4.2%)	10 (100%) 0 (0%)	/
Dialysis vintage (months)	37.8 [13.7–80.1]	21.8 [9.4–56.9]	19.4 [10.9–36.2]	0.217 ^c^
Necessity of hospitalization	16 (21.3%)	6 (25.0%)	5 (50.0%)	0.071 ^a^

^a^ Chi-square; ^b^ One-way ANOVA; ^c^ Mann-Whitney U test. ^§^ Significant A vs. C (*p* = 0.0008) and B vs. C (*p* = 0.00101) with pairwise comparisons by Tukey’s test.

## Data Availability

The data presented in this study are available on request from the corresponding author.

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
