# Peer review of "Efficacy of SARS-CoV-2 Vaccination in Dialysis Patients: Epidemiological Analysis and Evaluation of the Clinical Progress"

_jcm, 2022, doi:10.3390/jcm11164723_

Round 1

Reviewer 1 Report

The authors provide a comprehensive overview of the factors influencing dialysis patients during the pandemic. They describe the benefits of COVID-19 vaccination and the proportion of unvaccinated dialysis patients at significantly increased risk of disease. The manuscript thus provides important information on the effectiveness of vaccination in this particular patient population. This is particularly important against the background of possible immunodeficiency due to chronic dialysis therapy. 

Author Response

The authors provide a comprehensive overview of the factors influencing dialysis patients during the pandemic. They describe the benefits of COVID-19 vaccination and the proportion of unvaccinated dialysis patients at significantly increased risk of disease. The manuscript thus provides important information on the effectiveness of vaccination in this particular patient population. This is particularly important against the background of possible immunodeficiency due to chronic dialysis therapy. 

We sincerely thank you for your positive feedback and for the time spent to revise our paper.

Reviewer 2 Report

The authors have presented compelling data regarding the efficacy of COVID-19 vaccination in a population which is at high risk for mortality from this disease. It is impressive that study cohort had a high vaccination rate. The study is important in presenting evidence based discussion when counseling dialysis patients regarding benefits of covid -19 vaccination. 

Author Response

The authors have presented compelling data regarding the efficacy of COVID-19 vaccination in a population which is at high risk for mortality from this disease. It is impressive that study cohort had a high vaccination rate. The study is important in presenting evidence based discussion when counseling dialysis patients regarding benefits of covid -19 vaccination. 

We sincerely thank you for your positive feedback and for the time spent to revise our paper.

Reviewer 3 Report

The authors refer to a clinically significant issue. COVID-19 is a serious problem in patients with renal failure, causing significant mortality in dialysis patients. Vaccinations gave hope for an improvement in prognosis in this group of patients. Therefore, it is very valuable to assess the real effectiveness of vaccinations in the population of dialysis patients. However, in my opinion, the authors did not provide enough clinical data, there is no information on the causes of renal failure, body weight and BMI, comorbid diabetes, cardiovascular diseases, etc. With such little knowledge about the described population of dialysis patients, it is difficult to draw unequivocal conclusions. If the authors fill in all additional data on the study group, then I propose to resend the completely revised article.

Author Response

The authors refer to a clinically significant issue. COVID-19 is a serious problem in patients with renal failure, causing significant mortality in dialysis patients. Vaccinations gave hope for an improvement in prognosis in this group of patients. Therefore, it is very valuable to assess the real effectiveness of vaccinations in the population of dialysis patients. However, in my opinion, the authors did not provide enough clinical data, there is no information on the causes of renal failure, body weight and BMI, comorbid diabetes, cardiovascular diseases, etc. With such little knowledge about the described population of dialysis patients, it is difficult to draw unequivocal conclusions. If the authors fill in all additional data on the study group, then I propose to resend the completely revised article.

Thanks for you reasonable and constructive criticism. We have now collected more clinical data and added them to Table 1 with the related explanation in the text (Result section 3.1 “Characteristics of the population of dialysis patients”, page 4, lines 169-176).

Reviewer 4 Report

Mosconi et al reported on the efficacy of SARS-CoV-2 vaccination in dialysis patients. They studied a large cohort of dialysis patients and compared it to the general population.

The authors provide interesting data but not very new 

Major bias are:

- the absence of clinical characteristics of the dialysis patients: only gender, age, and dialysis vintage are available. The dialysis population is not homogenous and it is well established that some dalysis patients have increased risk of infections (those ith a past history of transplantation for instance). Such data would be useful to better characterize the risks and make adequate comparisons

- Vaccine responses are not available and once again such information would be of interest

THese datas should increase the relevance of the paper

Author Response

Mosconi et al reported on the efficacy of SARS-CoV-2 vaccination in dialysis patients. They studied a large cohort of dialysis patients and compared it to the general population.

The authors provide interesting data but not very new 

Major bias are:

  • the absence of clinical characteristics of the dialysis patients: only gender, age, and dialysis vintage are available. The dialysis population is not homogenous and it is well established that some dalysis patients have increased risk of infections (those ith a past history of transplantation for instance). Such data would be useful to better characterize the risks and make adequate comparisons.

Thanks for you reasonable and constructive criticism. We have now collected more clinical data and added them to Table 1 with the related explanation in the text (Result section 3.1 “Characteristics of the population of dialysis patients”, page 4, lines 169-176).

  • Vaccine responses are not available and once again such information would be of interest.

Regarding the ability of dialysis patients to develop adequate response to COVID vaccination, we did not treat this matter here, because it was not the focus of our research in this phase. We have just published an article (currently online only at https://www.mdpi.com/1648-9144/58/7/893/htm), where we described in general a satisfying level of protection after a 2-dose vaccination cycle with mRNA vaccines in dialysis patients and renal transplant recipient recovered form COVID-19, with a maximum peak at 3 months after the second dose. The overall message emerging from this study from our groups, although on a small population, seems to suggest that, even in those patients with immune dysfunction, COVID-19 vaccination can provide a stronger protection against re-infection and COVID severe illness with respect to the immunity conferred by SARS-CoV-2 infection itself, but the combined effect of both can trigger a stronger immune response.

We have mentioned this point in the discussion section as a study limitation, and added our recent article as ref.58.

THese datas should increase the relevance of the paper.

Round 2

Reviewer 3 Report

The authors responded to my comments.